# Biopolymer from Water Kefir as a Potential Clean-Label Ingredient for Health Applications: Evaluation of New Properties

**DOI:** 10.3390/molecules27123895

**Published:** 2022-06-17

**Authors:** Monalisa de Alencar Lucena, Igor Frederico da Silveira Ramos, Maurycyo Silva Geronço, Ricardo de Araújo, Francisco Lopes da Silva Filho, Luís Manuel Lopes Rodrigues da Silva, Rayran Walter Ramos de Sousa, Paulo Michel Pinheiro Ferreira, Josy Anteveli Osajima, Edson Cavalcanti Silva-Filho, Márcia dos Santos Rizzo, Alessandra Braga Ribeiro, Marcilia Pinheiro da Costa

**Affiliations:** 1Materials Science and Engineering Graduate Program, Federal University of Piauí, Teresina 64049-550, PI, Brazil; monalisaa.lucena@gmail.com (M.d.A.L.); igorfrederico10@gmail.com (I.F.d.S.R.); maurycyosg@gmail.com (M.S.G.); ricardo.biomedicopi@gmail.com (R.d.A.); josyosajima@ufpi.edu.br (J.A.O.); edsonfilho@ufpi.edu.br (E.C.S.-F.); marciliapc@ufpi.edu.br (M.P.d.C.); 2College of Pharmacy, Federal University of Piauí, Teresina 64049-550, PI, Brazil; filho_126_@hotmail.com; 3CPIRN-UDI/IPG—Centro de Potencial e Inovação em Recursos Naturais, Unidade de Investigação para o Desenvolvimento do Interior do Instituto Politécnico da Guarda, 6300-559 Guarda, Portugal; luisfarmacognosia@gmail.com; 4Laboratory of Experimental Cancerology (LabCancer), Department of Biophysics and Physiology, Federal University of Piauí, Teresina 64049-550, PI, Brazil; rayran.ramos@hotmail.com (R.W.R.d.S.); pmpf@ufpi.edu.br (P.M.P.F.); 5Pharmaceutical Sciences Graduate Program, Federal University of Piauí, Teresina 64049-550, PI, Brazil; marciarizzo@ufpi.edu.br; 6CBQF—Centro de Biotecnologia e Química Fina—Laboratório Associado, Escola Superior de Biotecnologia, Universidade Católica Portuguesa, Rua Diogo Botelho 1327, 4169-005 Porto, Portugal

**Keywords:** polysaccharide, biopolymer, dextran, photostability, mucoadhesiveness, antimicrobial activity

## Abstract

The present work aimed to characterize the exopolysaccharide obtained from water kefir grains (EPSwk), a symbiotic association of probiotic microorganisms. New findings of the technological, mechanical, and biological properties of the sample were studied. The EPSwk polymer presented an Mw of 6.35 × 10^5^ Da. The biopolymer also showed microcrystalline structure and characteristic thermal stability with maximum thermal degradation at 250 °C. The analysis of the monosaccharides of the EPSwk by gas chromatography demonstrated that the material is composed of glucose units (98 mol%). Additionally, EPSwk exhibited excellent emulsifying properties, film-forming ability, a low photodegradation rate (3.8%), and good mucoadhesive properties (adhesion Fmax of 1.065 N). EPSwk presented cytocompatibility and antibacterial activity against Escherichia coli and Staphylococcus aureus. The results of this study expand the potential application of the exopolysaccharide from water kefir as a potential clean-label raw material for pharmaceutical, biomedical, and cosmetic applications.

## 1. Introduction

Kefir grains consist of a symbiotic association of probiotic microorganisms surrounded by a polysaccharide matrix [1]. Water kefir grains are commonly used as a starter culture for fermentation, with sugar dissolved in water as the main substrate [2]. These grains are translucent, have a brittle structure, and are insoluble in water [3]. The microorganisms that compose the Brazilian water kefir grains have been described as a stable multi-species community consisting essentially of the lactic acid bacteria (LAB) group (~58%), primarily the genera Lactobacillus, Lactococcus, and Leuconostoc. In addition to these microorganisms, the grains are composed of the acetic acid bacteria (AAB) group (~31%), bifidobacteria, and yeasts, mainly those of the genera Saccharomyces (~11%) [4,5]. However, the predominant components of microbiota can be altered depending on the origin of the grain, substrates, and production methods that affect the diversity of the bacterial species that are present in the grains [5]. 

These different microorganisms have symbiotic relationships and use their bioproducts as sources of energy and growth for survival and multiplication [6]. They are used to produce a fermented drink which has been widely consumed for many years and has gained prominence due to its nutritional value and health benefits. Studies about the biological activity of kefir indicate potential anti-inflammatory, immunomodulatory, and antimicrobial activity, as well as anti-mutagenic and anticancer properties [2,7].

These grains are also responsible for producing exopolysaccharides (EPSs), classified as carbohydrate polymers that provide cell adhesion and protect microorganisms against adverse environmental conditions [8,9]. The EPS from water kefir grains is mainly constituted of dextran [10]. Dextran is the main component of the polysaccharide matrix of water kefir grains, produced by dextransucrase from sucrose, with consecutive α-(1→6) bonds in their main chains, which generally constitute 50% of the total bonds of this biopolymer. Dextran can be employed as a thickener, emulsifier, viscosity modifier, and stabilizer and is biodegradable [11,12].

The search for a replacement for petroleum-derived polymeric materials has valued the use of bio-based materials, such as biopolymers. Many foods or food residues have been a successful and sustainable alternative for obtaining biopolymers [13,14]. The market for natural polymers is increasing, especially to fulfill the growing demand for clean-label products with applicability in the food, cosmetic, and pharmaceutical industries [15]. EPSs of microbial origin are important natural alternatives since they present, in addition to technological benefits, numerous beneficial effects due to their antioxidant, antimicrobial, immunomodulatory, and anti-tumor properties, not found in more traditional plant-based polymers [8,16]. Considering these biological properties, EPSs are important alternatives in the biomedical area, such as in the development of films or hydrogels for application in wound healing and polymeric micro-and nanocarrier matrices for controlled drug delivery systems [17,18,19].

EPSs are valuable materials that do not have the environmental disadvantages associated with synthetic polymers, making them a more advantageous alternative to obtain on a large scale. The cultivation of grains is easier, cheaper, and requires less time than the process of isolating other microorganisms in microbiological culture media to produce EPSs [20].

The characterization tests carried out in this work aimed to show the potential of this biopolymer and expand its applications. Polysaccharides have been extensively studied regarding their functional properties. Functional properties evaluate the functionality of the materials and influence their applications [21,22]. A photostability assay assesses the protection of a biopolymer against UV light and can be important for its use as an encapsulant material of photosensitive substances. Another relevant property of biopolymers is mucoadhesiveness. Mucoadhesive biopolymers can improve the controlled release of drug delivery systems in the organism. The cyto- and hemocompatibility of biopolymers are important for biomedical applications of colloidal dispersions, hydrogels, films, or nanoformulations [23]. Thus, due to the necessity of exploring innovative clean label ingredients, the present study aims to characterize the exopolysaccharide (EPS) extracted from water kefir grains and evaluate their functional, technological, and biological properties for further applications, especially for products in the pharmaceutical, biomedical, and cosmetic areas.

## 2. Results and Discussion

### 2.1. Evaluation of the Growth Rate of Water Kefir Grains (GGR)

The results showed that the grain growth rate remained in the range of 85.02 ± 0.06% to 125.88 ± 0.13% (Appendix A). There was a statistically significant difference between the grain growth of cultivation one and the other cultivations. However, no statistically significant difference was observed between the grain growth of cultivations two, three, and four. The GGR obtained was higher than that observed in other studies that also used brown sugar as a substrate. Nascimento et al. [24] showed a water kefir grain biomass growth of approximately 77%. This result indicated that the fermentation conditions used in this study were favorable to maintain the viability of the grains and facilitate their growth. In addition, the growth rate of the grains increased with the cultivation progress conducted (Appendix A), suggesting an adaptation of the grains to the cultivation medium. These results directly influenced the production of the EPS because the grains are the source of this biomaterial, which is the object of the present study.

### 2.2. Production and Purification of EPSwk

A yield of 18.0 ± 8.4 mg g^−1^ (1.80%) EPSwk was obtained after the extraction process. Previous studies obtained the EPS from the isolation of microorganisms in culture media, thus obtaining a polysaccharide yield in g L^−1^. Li et al. [25] extracted the EPS from Leuconostoc mesenteroides strains isolated in culture media and obtained an EPS yield of 2.42 g L^−1^. Zhao et al. [26] obtained an EPS yield produced by Weissella confusa XG-3 of 97.5 ± 1.1 g L^−1^, and Rao and Goyal [27] obtained a yield of 38 g L^−1^ from the EPS produced by Weissella cibaria JAG8. The EPS yield varies when obtained from different strains since the production of EPSs by these microorganisms depends on the substrate composition and the environmental conditions used in the cultivation, as well as on the extraction method employed to obtain and purify the biomaterial [28]. 

### 2.3. Cryoprotection of Water Kefir Grains

Our objective in this trial was to verify whether the water kefir grains’ viability would be affected during the freeze-thaw process after using a cryoprotectant. Appendix A shows the results obtained in the evaluation of the GGR after 30 days of freezing. The growth rate of frozen grains without the addition of glycerol (control group) was significantly lower than the others, showing a gradual decrease in viability and growth rate for successive cultures. This result was similar, but with some distinct differences, to that obtained by Laureys et al. [3], one of the only studies addressing the so-called industrial water kefir production process. The authors stored the kefir grains at −20 °C with no cryoprotectant, and after thawing, the grain growth remained null in all subsequent fermentation processes. For the authors, the possible causes of the low grain growth and the slow progression of fermentation would be caused, respectively, by the use of demineralized water to produce kefir (low buffering capacity) and by the high concentration of sucrose (approx. 25% (*w*/*v*)), associated with the effect of osmotic pressure on microorganisms. On the other hand, our study used mineral water and brown sugar as a substrate at a 7% (*w*/*v*) concentration for all experimental groups, thus minimizing the osmotic stress on the microorganisms. Conversely, the groups with the cryoprotectant achieved significant growth of the grains in the three concentrations tested. The GGR varied from 37.93 to 131.18%, 100.92 to 142.33%, and 98.4 to 141.75% for glycerol concentrations at 10%, 25%, and 50%, respectively (Appendix A). In cultivation one, there was no statistically significant difference between the control and 10% of glycerol and between 25 and 50% of glycerol. In cultivation two, a statistically significant difference was not observed between the concentrations of 10 and 25% of glycerol and between 25 and 50% of glycerol. In cultivation three, there was no statistically significant difference between the concentrations of 10 and 25% of glycerol, 10 and 50% of glycerol, and between 25 and 50% of glycerol. In cultivation four, no statistically significant difference was seen between the concentrations of 25 and 50% of glycerol. The most significant grain growth occurred in cultivation three with glycerol at a concentration of 25%. 

The data confirmed that additives such as glycerol, DMSO, or sucrose had a possible effect on the intracellular microenvironment, maintaining the viability and metabolic activity of the microorganisms present in the kefir grains after freezing [29]. On cooling, depending on cell type and growth conditions, the phospholipid bilayer changes from a fluid and disordered liquid-crystalline phase to a rigidly ordered gel phase. Shortly after the membrane lipid phase transition, water crystallizes, and ice gradually forms in the extracellular matrix (ECM). Meanwhile, the ECM becomes progressively concentrated and exerts increasing osmotic pressure on the still not completely frozen intracellular compartment. The osmotic gradient removes water from the cells, promoting an intracellular concentration of solutes, decreasing the diffusion rates of intracellular molecules and an excess concentration of macromolecules. Cellular dehydration and progressive reduction in volume lead to a glassy state in the cytoplasm, in which high viscosity severely restricts the diffusion of oxygen and metabolites, and the cell passes into an inactive state. Thus, in the absence of additives, intracellular vitrification temperature is observed early, and the effect of the additives is to fluidize the intracellular environment, limiting freeze-thaw cellular damage [29,30].

Regardless of the geographic origin of the grains and the fermentation substrate, water kefir grains were present as microbial members of their primary composition in lactic acid bacteria (LAB) of the genus Lactobacillus, acetic acid bacteria of the genus Acetobacter, and yeasts of the genus Saccharomyces [5,10]. Many studies indicate that these three main strains are important for the elaboration of fermented beverages, with LAB being considered key members in EPS production. Therefore, additives for freezing water kefir cultures reinforce the potential of using kefir grains on an industrial scale to obtain EPS and subsequent application in the biomedical and pharmaceutical industries.

### 2.4. Characterization of EPSwk

#### 2.4.1. Physico-Chemical Parameters and Zeta Potential (ζ)

Appendix A presents the pH values of the EPS solutions at different concentrations. A slight increase in the values was observed with the increase in the concentration of the EPS. However, the pH remained slightly basic in all concentrations, differing from the literature, in which dextran exhibited neutral pH [31]. This result may be related to the extraction process that uses a basic solution (sodium hydroxide, NaOH) in one of the steps, suggesting that the neutralization stage was not completely efficient. Thus, an alternative would be to carry out the dialysis membrane filtration process to improve the purification and remove possible residues from the basic solution [32]. Despite this, the pH values found for EPSwk solutions can be easily adjusted to the specific pH according to the desired application for the biomaterial since normal skin presents a pH of 5.5, soft tissues and acneic skin present a pH of around 7.4, and atopic skin a pH of around 8.5 [33].

The results obtained in the analysis of the electrical conductivity of EPSwk were 155.00 ± 20 μS/cm (0.5%), 350.00 ± 28 μS/cm (1.0%), 512.00 ± 30 μS/cm (2.0%), and 691.50 ± 15 μS/cm (3.0%) (Appendix A). The conductivity increased proportionally to the increase in the concentration of the biomaterial solution. This result was due to the increase in the number of free ions and, consequently, increased mobility in the aqueous medium, as described by the free ion theory [34]. Ponrasu et al. [35] reported a conductivity of 27.5 µS/cm for 10% pullulan, an EPS produced by the polymorphic fungus Aureobasidium pullulans. EPSwk showed in lower concentrations showed better conductivity values than the pullulan. The high conductivity value of EPSwk when compared to the scientific literature, may be related to the presence of residual NaOH in the extraction step. This result suggests that the EPSwk has excellent potential for use in the production of hydrogels intended for application in the neural regeneration process since it presented ideal conductivity values (between 0.1 and 1000 μS/cm) to stimulate cellular activities, such as proliferation and differentiation, thus helping to restore the conductive neural pathways that were interrupted [36]. This biopolymer can also be used as a carrier for electro-responsive drugs because electrically conductive polymeric materials can undergo oxidation or reduction processes, which allows the controlled release of drug molecules incorporated in their structure [37].

The zeta potential (ZP) analysis was performed to obtain information on the surface load of EPSwk. The ZP value for EPSwk was −10.3 mV, indicating that the material has an anionic character, which may be related to the presence of hydroxyl ions (OH^−^) adsorbed on the surface of the biopolymer remaining from the alkaline extraction method [38]. In previous studies, a ZP of −4.35 mV was obtained for the EPS produced by Lactobacillus helveticus MB2-1 [39], and −13.9 mV and −12.1 mV for the EPS produced by two strains of Nostoc sp. [40]. In another study, the EPS obtained from Aeromonas punctata showed a positive charge up to pH 5, and from that pH value, there was a change to a negative charge, with a ZP of approximately −20 mV at pH 8 [41]. This result was due to the direct influence of pH on the zeta potential values of a given material, making it positively (low pH) or negatively (high pH) charged [42]. ZP is one of the most relevant parameters to evaluate the electrokinetic potential in colloidal systems. ZP reveals the presence of surface charges, which can influence interaction forces among electrically charged particles in a dispersion [43,44,45]. In this low ZP, EPSwk forms a colloidal suspension considered physically unstable [44]. A high ZP (positive or negative) indicates the potential stability of the colloidal system. Generally, a zeta potential value greater than +30 mV or lower than −30 mV has sufficient repulsive force to reach better colloidal stability. On the contrary, when the particles have low ZP values, they will not have sufficient repulsive force to prevent aggregation or flocculation [44,45]. Thus, the surface charges determined by the zeta potential can influence several properties of colloidal dispersions, among them, the stability, flocculation, and aggregation of colloidal systems, the electrophoretic mobility of the suspended particles, the formation of complex coacervation, the swelling of the polymers, and the mucoadhesive properties of a polymer [44,46,47,48,49].

#### 2.4.2. Determination of Molecular Weight (Mw)

EPSwk presented an Mw of 6.31 × 10^5^ Da, calculated from the standard curve. This result complies with the literature because for polysaccharide materials, including EPSs of microbial origin, the Mw values vary from 10^4^–10^6^ Da [50]. According to Mnif and Ghribi [51], EPSs are high molecular weight polysaccharides. In comparison with previous studies, EPSwk showed an Mw similar to the EPS obtained from L. mesenteroides BD1710 (6.35 × 10^5^ Da) [52] and a value close to the dextran obtained from Leuconostoc pseudomesenteroides YB-2 (7.67 × 10^5^ Da) [53] and the EPS produced by Sporidiobolus pararoseus JD-2 (7.4 × 10^5^ Da) [54].

The Mw of dextran differs according to the source of production. This characteristic reflects in the properties and application of this biomaterial. Polysaccharides with low Mw have better solubility in water, improving their bioactivity and facilitating their application in biological systems for carrying drugs and bioactive compounds [55]. High Mw dextrans are used to improve the texture of foods and exhibit excellent potential for modifying the rheological properties of formulations [56]. Thus, molecular weight is an important parameter that influences the application of biopolymers in several sectors, such as the pharmaceutical, food, cosmetic, and chemical industries [57].

#### 2.4.3. Monosaccharide Content

A chromatographic analysis by gas chromatography of the monosaccharides from water kefir grains can be seen in Figure 1. This result demonstrated that the material consists of a homopolysaccharide composed of glucose units (98 Mol%), confirmed by the only peak presented in the chromatogram, which corresponds to this monosaccharide [57,58,59,60]. This single chromatographic peak is comparable to that reported in a previous study for exopolysaccharides produced by Bacillus anthracis strain PFAB2 [61]. Furthermore, this result indicates that the EPS obtained from the water kefir grains is dextran, as found in the studies by Zhou et al. [62] and Hashem et al. [63], which demonstrated that dextran consists of a long-chain biopolymer of D-glucose. The presence of only glucose as the main monosaccharide component can influence the structural and functional properties of this biopolymer [64].

#### 2.4.4. Protein Content

The analysis of the Dumas method found that the EPSwk protein content was only 0.57%, which indicated the effectiveness of the purification stage of the biomaterial extraction process, as the content of proteins is directly related to the degree of purity of the exopolysaccharide [65]. Previous studies have also reported low protein content in polysaccharides of microbial origin, with 1.8% EPS obtained from W. confusa 11GU^−1^ [66] and 0.49% EPS extracted from Oenococcus oeni [67].

#### 2.4.5. Fourier Transform Infrared Spectroscopy (FTIR)

The functional groups present in the EPSwk were investigated with FTIR spectroscopy. The results illustrated in the FTIR spectrum (Figure 2a) confirmed that the analyzed material is a polysaccharide due to the presence of absorption bands in five specific regions: region Ⅰ (4000–2500 cm^−1^), region Ⅱ (1800–1500 cm^−1^), region III (1500–1200 cm^−1^), region Ⅳ (1200–800 cm^−1^), and region Ⅴ (below 800 cm^−1^) [68]. The high-intensity band at 3371 cm^−1^ corresponds to O─H stretching vibrations [62]. The bands observed at 2935 cm^−1^ and 2906 cm^−1^ are attributed to C─H stretching vibrations [69]. Absorption at 1649 cm^−1^ corresponds to the stretching vibrations of the C=O bond [70]. The absorption band at 1420 cm^−1^ and 1346 cm^−1^ was related to C─O stretching vibrations [71] and C─H deformation vibration [72], respectively. The presence of absorption at 1152 cm^−1^ and 1016 cm^−1^ are characteristic of C─O─C vibrations and the presence of α-(1→6) glycosidic linkages [49], respectively. The band at 918 cm^−1^ indicated the asymmetric vibrations of the glucopyranose ring [73]. The small band at 767 cm^−1^ was related to C─C bonds [72]. Thus, the profile of functional groups and bonds presented in the FTIR spectra is an excellent indicator that the EPSwk corresponds to purified dextran. 

#### 2.4.6. X-ray Diffraction (XRD)

X-ray diffraction (XRD) was used to evaluate the crystallinity of the EPS sample. The observed diffraction profile (Figure 2b) indicated that the EPSwk had an amorphous structure since the diffractogram did not contain well-defined peaks. Such characteristics in polysaccharide diffractograms suggest the polymeric nature of the EPS, as observed in a previous study [74]. Furthermore, the diffractogram showed the formation of a broad peak at 2θ° in the range of 15–25°, which is also observed in the analysis of the crystallinity profile of commercial dextran [75] and dextran produced by strains of Lactobacillus mali CUPV271 and Leuconostoc carnosum CUPV411 [76].

#### 2.4.7. Thermal Properties

A thermogravimetric analysis is a simple and accurate method used to verify the thermal stability and the decomposition pattern of a given material [77]. The TGA and DTG (Derivative Thermogravimetry) curves corresponding to the EPSwk are illustrated in Figure 2c. The curve showed that the EPS tends to degrade in three stages. In the first stage, starting at 50 °C, a small initial loss of mass (≅10%) was observed up to 200 °C, which is related to the loss of free water linked to the material structures, and from the condensation of hydroxyl groups on the surface [78]. As the temperature increased, there was a second stage of considerable mass loss (≅50%), between 220 °C and 300 °C, indicating the occurrence of the EPS depolymerization process by breaking C─C and C─O bonds present in sugar rings and the release of small molecules from this depolymerization, as well as the discharge of waste water from the condensation of the innermost hydroxyl groups [79]. 

The TGA curve indicated that the EPSwk suffered maximum degradation at a temperature close to 250 °C, confirmed by the degradation peak observed in the DTG curve (≅270 °C); this result was similar to that observed by Sajna et al. [80]. In the third and last stage, starting at 400 °C, the mass loss gradually decreased. Moreover, a 25% residue of the total mass was obtained at a temperature of 800 °C [81]. The EPS thermogravimetric profile of EPSwk suggests that this material has similar thermal stability to carbohydrate polymers. This property is important for applications of this biopolymer in the biomedical area, in the pharmaceutical, cosmetic, or food industries, and in formulations or products that must remain stable at the high temperatures used in the manufacturing and processing, which guarantees greater stability and greater added value to the final product [82]. In addition, processing temperatures are a critical factor in the pharmaceutical industry because they can degrade inputs or drugs. In this way, EPSwk can remain thermally stable during processes usually used industrially, including sterilization by autoclavation (15 min at 121 °C) [83], spray dry processing (<200 °C) [84], granulation and drying by Fluid Bed Dryer (<200 °C) [85], among others.

Differential scanning calorimetry (DSC) evaluates the thermal transitions or crystalline melting of polysaccharides [77]. Figure 2c illustrates the DSC thermogram of EPSwk performed with heat flow from 25 to 800 °C. The result showed an endothermic peak at 264.99 °C, which was possibly due to the degradation of the biopolymer [79]. In general, the endothermic peak is related to the breaking of hydrogen bonds and the loss of hydroxyl groups when the material is warmed. Consequently, the formation of the endothermic peak is because of the disruption of the structure [86].

As reported by Zhang and Chu [87], commercial dextran presented a high glass transition (Tg) of 222.7 °C and a non-crystalline state. A possible explanation for this high Tg would be the presence of strong hydrogen bonding among the macromolecules of dextran. As previously reported, some dextrans have presented high degradation temperatures indicating no crystalline state and suggesting an amorphous behavior [76,87]. Previous studies have shown high Tg values for dextrans produced by lactic acid bacteria, among them Leuconostoc pseudomesenteroides (278.36 °C) [88], Lactobacillus mali CUPV271 (226.8 °C), and Leuconostoc carnosum CUPV411 (225.7 °C) [89]. According to Jiang et al. [90], the thermal behavior of the EPS from diverse lactic acid bacteria strains might differ. These temperature differences might be attributed to the different molecular configurations of the biopolymers [79].

#### 2.4.8. Scanning Electron Microscopy (SEM)

Figure 3 is the electron micrograph of the EPSwk. This method allowed precise morphological analysis of the three-dimensional structure and the morphology of the EPS surface, which contributes to the study of its functional, sensory, and physical properties [91]. The micrographs illustrated that the EPS surface was rough and irregular, with few pores (Figure 3), a result similar to a previous study [92]. The shape and surface characteristics may have been impacted by the extraction and purification methods used. In addition, the irregular distribution of the biopolymer particles can increase hydration capacity by increasing surface area [93]. These characteristics allow this material to be used as a release retarding agent in controlled drug delivery systems, as previously reported in other polysaccharide materials [94]. 

### 2.5. Functional Properties 

The results of the functional properties of EPSwk are provided in Appendix A. The water solubility of the EPS is one of the most important functional properties and depends on the polymer structure (length of the main and branched chains), the degree of polymerization, and the arrangement of the glycosidic bonds [88,95]. The EPS had moderate (53.7%) solubility, which was lower than the dextran produced by *L. pseudomesenteroides* (92%) [62]; however, that polymer solution was subjected to stirring and heating steps. The temperature factor influences the solubility of the material, as it reduces aggregation between the particles and promotes the transport of water to the polymer granules [96]. According to Ahmadi et al. [97], the interaction between hydrocolloids and water structures is affected by hydrogen bonding, temperature, and particle size. In addition, they mention that a heating process is required to completely dissolve hydrocolloids to achieve their whole viscosity in most cases. Thus, the EPSwk drying process may have influenced the formation of aggregates, which decreased the solubility of the biopolymer. This moderate solubility is a disadvantage because it would require more time and costs in the industrial processing of this material [97]. 

The water holding capacity (WHC) is an important property that evaluates the capacity of samples to absorb and hold water molecules [64,97]. The WHC was 704.7%, higher than the dextran produced by W. cibaria (287.84%) [98]. WHC is influenced by high molecular weight, conformational structure, and hydrophilic regions of the molecules. Other factors that can also influence the WHC are the capillarity of the molecules, the pore size, and the drying method of the samples [90,97]. This result demonstrates that the EPSwk has good hydrophilicity and excellent potential to maintain a high amount of water in its structure through hydrogen bonds, to avoid syneresis or be used as a viscosity modifier in pharmaceuticals or cosmetic formulations [99]. The emulsifying and foaming capacities reduce surface tension in emulsions, facilitate the oil-water interaction, and reduce the coalescence rate [100]. The EPSwk had an emulsifying ability similar (89.3%) to the dextran produced by *L. pseudomesenteroides* (87.22%) [88]. In addition, it presented good emulsifying stability (Appendix A) which allows the use of this material as an emulsifying agent in formulations. The result obtained for the foaming ability (26.7%) demonstrates that the material can be used as a stabilizer and thickener in formulations that do not require foam, both in the pharmaceutical and food industries. This factor can be considered an advantage over other materials since the formation of foam is related to the loss of quality in several formulations.

The polymer sample exhibited a relatively high swelling index value (43.1 g/g), which agrees with the results obtained for polysaccharides of microbial origin presented in the literature [101]. This result suggests that the EPSwk may perform well as a matrix, binding, or disintegrating agent [102]. In addition, the swelling capacity is an important factor in evaluating other properties of the biomaterial since a biopolymer with adequate swelling capacity may be able to obtain controlled and uniform drug release and promote effective mucoadhesive capacity [103].

Appendix A illustrates the results of the film of EPSwk. In this initial evaluation, the biopolymer seemed to be able to form a film with and without the addition of a plasticizer. This result was due to the entanglement process of the long EPS chains during the dehydration of the film-forming solution [78]. The possible film-form property makes this biomaterial an excellent candidate for several applications in the biomedical area since it can be used as a polymeric matrix for the production of controlled drug delivery systems and in the development of wound healing dressings. In addition to the adhesive properties, this material is biocompatible and biodegradable, which makes its use in these applications advantageous [78,104].

### 2.6. Photostability Study

Figure 4 charts the results of the photostability study of EPSwk. A reduction of 8.33% in the intrinsic viscosity of the EPS solution was observed after the maximum exposure time (120 min). A previous study demonstrated a reduction in intrinsic viscosity of ≅18% for oat starch and 30% for barley starch after 30 min under UV irradiation [105]. In comparison to the data obtained in the literature, the EPSwk presented a low rate of photodegradation of its polymeric chain, which is an important property and could greatly influence the application of this material in the biomedical area.

The photodegradation rate calculated for the EPS sample was 3.80% after the maximum exposure time (120 min) (Figure 4), corroborating the data obtained in the analysis of the intrinsic viscosity since this significantly low value demonstrates the excellent photostability of the analyzed material. The structural diversity of polysaccharides and the presence of chromophore groups capable of absorbing UV light in the polymeric chains influence the photostability property [106]. The presence of proteins in the sample can also influence this property because the main chromophores absorbing in the UV region are the aromatic amino acids, tryptophan, tyrosine, and phenylalanine [107]. The characteristics of the EPSwk polymer chains and the amount of proteins found in the EPSwk sample (0.57%) can have influenced the increase in photostability. Thus, this property makes the EPSwk an excellent alternative for application in the biomedical, cosmetic, and pharmaceutical industries in formulations incorporating bioactive and photosensitive compounds due to its ability to absorb UV light and to protect these molecules from the photodegradation process, as some of the pharmacologically important chemicals can have severe toxic reactions when exposed to UV radiation [108]. The scarcity of analysis on the photodegradation of EPS demonstrates the great importance of these results, as it highlights the great potential of this biopolymer in the production of new carrier systems for photosensitive bioactive compounds in formulations, contributing significantly to the advancement of the pharmaceutical industry for the use of these readily available biomaterials as well as its great technological and commercial potential [109].

### 2.7. Biological Assays

#### 2.7.1. Ex Vivo Mucoadhesiveness

Figure 5 shows the determination of mucoadhesiveness of EPS samples. The hydrogel presented an adhesion force (Fmax) of 0.040 ± 0.020 N and work of adhesion (Wad) of 0.015 ± 0.007 N.sec, and the powder sample presented a Fmax of 1.065 ± 0.035 N and Wad 0.605 ± 0.063 N.sec. The values of Fmax and Wad were significantly higher for the powder sample than for the hydrogel because the interaction of the polysaccharide chain with the mucosa is much more significant with the polymer in solid forms, such as powder, films, or tablets. The powder form contains more free groups (such as hydroxyls, responsible for the formation of hydrogen bonds with the components of the mucin) than the biopolymer in a fully hydrated form, such as solutions and hydrogels [110]. Excessive hydration results in a decrease in the mucoadhesiveness of the polymer due to the interaction of water molecules with polymeric chain groups responsible for adhesion to the mucosa [103].

Bassi da Silva et al. [111] observed that binary hydrogels consisting of Poloxamer 407, Carbopol 971P^®^, Carbopol 974P^®^, and Noveon^®^ Polycarbophil had lower mucoadhesive strength values than the EPSwk hydrogel. In another work, Hall et al. [112] found that hydrogels synthesized by copolymerization with 2-hydroxyethyl methacrylate (HEMA) and N-acryloyl glucosamine (AGA) had mucoadhesive strength values close (~0.05 N) to the EPSwk hydrogel.

Numerous factors are decisive for the mucoadhesive properties of polysaccharides, such as molecular weight, viscosity, concentration, pH, and other characteristics of the polymer chain, which involve length, conformation, degree of cross-linking, and flexibility [113,114]. The charge and degree of ionization of the polymer are also very important factors that influence the mucoadhesion strength, and anionic polymers have greater mucoadhesion strength [114]. Other important factors for the mucoadhesive property of EPS are its swelling capacity and high WHC. When applied to the mucosal surface, the EPS can absorb the water present in the mucus gel resulting in the interaction of the material with the mucin and providing greater adhesion [115]. Thus, the good mucoadhesive characteristics of EPSwk are justified by the anionic characteristic of this biopolymer, its high molecular weight, high degree of swelling, and water holding capacity. The mucoadhesive properties of this biopolymer are important for its application in the biomedical area, as it allows the development of mucoadhesive films (Figure 5). These films can be used in the controlled release of drugs in specific mucous membranes or the systemic release through the adhesion of the formulation in the tissues or cells present at the absorption site, thus configuring itself as an alternative to the treatment of numerous diseases [116].

#### 2.7.2. Antimicrobial Activity

The EPSwk exhibited antimicrobial activity against both tested bacteria, with a minimum inhibitory concentration (MIC) of 5000 µg mL^−1^ and 2500 µg mL^−1^ against S. aureus and *E. coli*, respectively. The Gram-negative bacteria (*E. coli*) were more susceptible to the presence of EPSwk than the Gram-positive bacteria (*S. aureus*). The result can be explained by the constitution of the cell wall of these microorganisms since Gram-positive bacteria cells have a thick layer of peptidoglycans, in addition to other secondary structures, such as teichoic acids and other polysaccharide components [116], which provide microorganisms greater protection against possible antimicrobial agents. 

This activity against *E. coli* (MIC = 2.0 mg mL^−1^) and S. aureus (MIC = 3.0 mg mL^−1^) was also observed by Ye et al. [43], who evaluated the antimicrobial activity of dextran obtained from *L. pseudomesenteroides* YB-2. Another study found that the EPS extracted from milk kefir grains (kefiran) also achieved greater antimicrobial activity against Gram-negative bacteria (*E. coli* and *Pseudomonas aeruginosa*) than Gram-positive bacteria (*S. aureus* and *Streptococcus faecalis*) [117]. Compared with these results, the antimicrobial activity of EPSwk is lower in both bacteria. This difference may be related to the types of microorganisms in kefir grains able to produce antimicrobial substances. According to the scientific literature, many bacteria of the Lactobacilaceae family produce antimicrobial peptides such as bacteriocins [118].

#### 2.7.3. In Vitro Cytotoxicity Assay

Caco-2 cell line is commonly used to assess the intrinsic cytotoxicity of new materials, drugs, or drug delivery systems [119,120,121]. The results obtained for the in vitro cytotoxicity analysis of the EPSwk are in Figure 6. There was no statistically significant difference (*p* > 0.05) in cell viability, compared to the control group, after treatment with the EPS in all tested concentrations. This result was similar to Shahnaz et al. [122], who evaluated the cytotoxicity of the carboxymethyl dextran polymer using the MTT test in Caco-2 cells obtaining cell viability of 99.5 ± 0.1%. Another study found a similar result, in which a polysaccharide isolated from mushrooms (β-D-glucan) did not exhibit toxicity in Caco-2 cells until the concentration of 1000 µg mL^−1^ [123]. Therefore, the high rate of cell viability proved that EPSwk is not cytotoxic, which reinforces and enables its application in oral administration systems [124] in the form of mucoadhesive films, as previously proposed.

#### 2.7.4. Hemocompatibility Assay

Appendix A illustrates the results of the hemocompatibility evaluation test of EPSwk. The EPSwk (1000 µg mL^−1^) did not form a hemolytic halo in the Blood Agar (Appendix A), which indicated that this biomaterial is hemocompatible, corroborating the results obtained in the cytotoxicity analysis.

Hemolysis assays are important for assessing the hemocompatibility of biomaterials that are intended to be applied in biological systems and may come into contact with the blood since damage to erythrocytes results in the release of hemoglobin, and this is considered the first evidence of cytotoxicity in the human body [125]. For this reason, the analysis of the hemocompatibility of these biomaterials is necessary for their effective and safe application in biological systems. This analysis differentiates this study because previous studies rarely conducted hemolysis assays for exopolysaccharide materials.

#### 2.7.5. Microbiological Control and Water Activity (Aw)

Microbiological control of EPSwk was conducted to ensure the efficiency of the polymer extraction process and guarantee the microbiological quality and safety of the material to consequently allow it to be used safely as a functional component in several applications. The Aw of the EPSwk was 0.54 ± 0.01, a result higher than that obtained by Días-Montes et al. [126] (in the range of 0.2 to 0.3), which evaluated the water activity of dextran obtained from L. mesenteroides SF3. However, the result obtained is within the range where the probability of microbial growth in the material is low since microorganisms need water activity between 0.6 and 0.8 to develop their metabolic and physiological activities [126,127]. This indicates that the material is stable in the face of microbial deterioration and reinforces its application potential.

## 3. Materials and Methods

### 3.1. Materials

The kefir grains were obtained from a local store (Teresina, Piauí, Brazil) and were registered with the Brazilian National System of Management of Genetic Heritage and Associated Traditional Knowledge (SisGen) under No. A79599F. Sodium hydroxide, hydrochloric acid, anhydrous glycerol (glycerin), and absolute alcohol were analytical grade. Standard kit P-82 Shodex pullulan (Model WAT034207), composed of P-800 (736 kDa), P-400 (348 kDa), P-200 (200 kDa), P-100 (113 kDa), P-50 (49 kDa), P-20 (23 kDa), P^−^10 (9.9 kDa), and P-5 (6.6 kDa), was purchased from Waters™ (Madrid, Spain).

### 3.2. Evaluation of the Growth Rate (GGR) of Water Kefir Grains 

The manufacture of kefir grains was adapted from the method described by [128]. Artisanal culture of the kefir (50.0 g) was subjected to four consecutive and independent cultivations (cultivations 1, 2, 3, and 4), using brown sugar as a substrate in the proportion of 500.0 mL: 35.0 g (mineral water: brown sugar), for a cultivation period of 2 days at a temperature of 25 °C. Then, water kefir grains obtained in cultivation 1 were used in cultivation 2; the grains obtained in cultivation 2 were used in cultivation 3, and the grains obtained in cultivation 3 were used in cultivation 4. The grain growth rate (GGR) was calculated according to Equation (S1). This analysis was performed in triplicate.

### 3.3. Extraction of Exopolysaccharide from Water Kefir Grains (EPSwk)

The exopolysaccharide from water kefir grains (EPSwk) from cultivation 4 was extracted according to previously published methods [129,130], with adaptations. A mixture of grains and distilled water in a proportion of 1:10 (*w*/*v*) was prepared, which was subjected to heating (80 °C) and constant stirring (680 rpm) for 40 min. Subsequently, the mixture was centrifuged at 4000 rpm for 10 min (Digital Centrifuge TC-Spinplus-8, Spinplus™), and the precipitate was placed in a beaker containing NaOH solution (0.1 mol L^−1^) in the proportion of 1:2 (*w*/*v*), and the initial step of heating (80 °C) and stirring (680 rpm) for 40 min was repeated. The mixture was cooled, filtered, and the obtained solution was submitted to neutralization (pH = 7) with HCl solution (0.1 mol L^−1^). It was precipitated in absolute ethanol in the proportion of 1:3 (*v*/*v*) (overnight). The material was isolated from alcohol by centrifugation (4000 rpm for 15 min), and the precipitate was dried at 40 °C for 24 h.

### 3.4. Cryoprotection of Water Kefir Grains

For cryoprotection, 20.0 g of water kefir grains were subjected to the freezing process with the addition of glycerol in concentrations of 10%, 25%, and 50% (*m*/*m*) and without glycerol (control) for 30 days. Then, water kefir grains obtained in cultivation 1 were used in cultivation 2; the grains obtained in cultivation 2 were used in cultivation 3; and the grains obtained in cultivation 3 were used in cultivation 4. After the freezing period, the grains were cultivated in water and brown sugar (200.0 mL: 14.0 g) to evaluate the grain growth rate (GGR). The analysis was performed in triplicate.

### 3.5. Characterization of EPSwk

#### 3.5.1. Physico-Chemical Parameters and Zeta Potential (ζ)

Polymer solutions in concentrations of 0.5%, 1.0%, 2.0%, and 3.0% (*w*/*v*) were prepared for pH and conductivity assessment, with a digital pH meter and a pocket conductivity meter, respectively. The distilled water used to prepare the polymer solutions was used as a control. The analysis was performed in triplicate. The Zeta Potential (ζ) was measured in Zetasizer Nano-ZS90 (Malvern™, Worcestershire, UK) at 25 °C for dispersion of EPSwk at a concentration of 0.1% (*w*/*v*) prepared in purified water (Milli-Q System Millipore, Bedford, MA, USA), readings were taken in triplicate after five minutes of equilibration.

#### 3.5.2. Molecular Weight

The molecular weight (Mw) of EPSwk was determined using chromatographic analyzes carried out in an Agilent™ 1260 Infinity II LC System chromatograph, equipped with Agilent™ 1290 Infinity II Evaporative Light Scattering Detector (ELSD), with N2 flow of 1.2 slm, an evaporator temperature of 70 °C and a nebulizer temperature of 50 °C. PL Aquagel-OH MIXED-M columns (part number: PL1149-6801, 4.6 × 250 mm, 8 µm) and Aquagel-OH 20 (part number: PL1120-6520, 300 × 7.5 mm, 5 µm) were used as the stationary phase, with a pre-column of PL Aquagel-OH (part number: PL1149^−1^240, 7.5 × 50 mm, 15 µm). Ammonium acetate solution (10 mmol L^−1^) was used as a mobile phase at a flow rate of 0.6 mL.min^−1^. The sample was prepared in ammonium acetate solution (10 mmol L^−1^) and filtered with a 0.22 μm syringe filter to ensure homogeneity. The system was calibrated with the standard pullulan kit of varying molecular weights (736 kDa P-800; 348 kDa P-400; 200 kDa P-200; 113 kDa P-100; 49 kDa P-50; 23 kDa P-20; 9.9 P-10 kDa; and 6.6 kDa P-5). All chromatograms were analyzed using OpenLab ChemStation software (Agilent™, Santa Clara, CA, USA).

#### 3.5.3. Protein Content

The protein content was determined by the Dumas method [65]. About 150 mg of sample was weighed in an aluminum crucible and analyzed on the Dumatec™ 8000/FOSS equipment (He flow rate: 195.0 mL min^−1^ and O2: 400 mL min^−1^, pressure: 1200 mBar). The protein content was determined from the total nitrogen content, multiplied by the conversion factor of 6.25. Ethylenediaminetetraacetic acid (EDTA) was used as a standard for constructing the nitrogen calibration curve (10.9 mg–150.2 mg).

#### 3.5.4. Analysis of the Monosaccharide Composition

The methodology analysis of the monosaccharides of the EPSwk was described by Coelho et al. [131]. The sample was subjected to prehydrolysis in 0.2 mL of 72% H_2_SO_4_ (*w*/*v*) for 3 h at 25 °C, followed by hydrolysis for 2.5 h in 1 mol L^−1^ H_2_SO_4_ at 100 °C. An Agilent 7890B Gas Chromatograph (GC) System equipped (Agilent Technologies, Santa Clara, CA, USA) with split/splitless capillary inlet and flame ionization detector (FID) was used in this work. The column was a DB-225 capillary column with 30 m length, 0.25 mm diameter, and 0.15 µm thickness. The internal standard employed was 2-deoxyglucose.

#### 3.5.5. Fourier Transform Infrared Spectroscopy (FTIR)

The spectra were obtained on the Spectrum™ 400 spectrophotometer (PerkinElmer, Waltham, MA, USA) using KBr tablets. KBr tablets (diameter, 5 mm) containing 500 mg dried KBr powder and 5 mg sample were pressed using a bench punch machine. The background spectrum was measured against a pure KBr tablet. The spectra of the exopolysaccharide samples were recorded in the range of 4000–500 cm^−1^, with 64 scans and a spectral resolution of 4 cm^−1^. The spectrum was obtained and treated using the OriginLab^®^2021b software. After plotting the graph, adjustments were made to the scales of the axes. Legends were added, and the graph was exported in TIFF format.

#### 3.5.6. X-ray Diffraction (XRD)

The analyzes were conducted at room temperature in an XRD-6000 device (SHIMADZU™, Kyoto, Japan), using copper Kα radiation (1.5418 Å), 40 kV voltage, and 30 mA current. The EPSwk sample was examined at a 2θ angle ranging from 5.0 to 75.0 degrees at a speed of 2° min^−1^.

#### 3.5.7. Thermogravimetry (TGA) and Differential Scanning Calorimetry (DSC)

The thermogravimetric analysis was performed using a DSC-TGA thermogravimetric analyzer (SDT Q600 V20.9) under an argon atmosphere (100 mL min^−1^). For the scan, the samples were weighed in alumina pans (mass of 5 mg) and heated in a temperature range of 25–800 °C at 10 °C min^−1^.

#### 3.5.8. Scanning Electron Microscopy (SEM)

The morphological characteristics of the EPSwk were performed by scanning electron microscopy (SEM) through an electron microscope Quanta FEG 250—FEI™ (FEI Company, Hillsboro, OR, USA) with an acceleration voltage from 1 to 30 kV. The sample was fixed on aluminum support and covered with gold (Au) in a Q150R metallizer (Quorum™, Lewis, UK) for 30 s at 20 mA, by plasma generated in an argon atmosphere. The sample images were recorded digitally in variable magnifications (1000 to 50,000×).

### 3.6. Functional Properties

The analysis of the functional properties of the EPSwk was carried out according to methodologies described by Alpizar-Reyes et al. [96], with adaptations. All analyses were performed in triplicate.

#### 3.6.1. Emulsifying Ability (EA) and Emulsifying Stability (ES)

Initially, 10 mL of a 3% (*w*/*v*) solution of EPSwk was prepared in distilled water. The emulsions were prepared by homogenizing 10 mL of a 3% (*w*/*v*) solution of EPSwk with 2.5 mL of soybean oil, maintaining the 1:4 ratio of oil: solution (*v*/*v*), with an Ultra-Turrax T50 homogenizer at 6600 rpm for 3 min. The emulsifying ability was calculated by Equation (S2). After the homogenization step, the emulsion remained at rest for 30 min. The emulsion was then centrifuged for 10 min at 1600 rpm. The emulsified layer was measured, and the emulsion stability (ES) was calculated by Equation (S3).

#### 3.6.2. Water Holding Capacity (WHC)

Dispersion of 3% (*w*/*v*) of EPSwk was prepared in distilled water and placed in previously weighed centrifuge tubes. Then, the dispersion was centrifuged for 15 min at 1600 rpm. The supernatant was discarded, and the sample was reweighed. The WHC was calculated by Equation (S4).

#### 3.6.3. Water Solubility (Sol)

Dispersion of 3% (*w*/*v*) was prepared with 0.3 g of dry EPSwk in 10 mL of distilled water. The dispersion was centrifuged for 15 min at 1600 rpm. Mnif. The solubility was calculated using Equation (S5).

#### 3.6.4. Foaming Ability (FA) and Foam Stability (FS)

Dispersion of 3% (*w*/*v*) of the exopolysaccharide was prepared. Then, the solution was subjected to stirring in Ultra-Turrax T50 at 6600 rpm for 5 min. The foaming capacity was calculated immediately after stirring (≅30 s), according to Equation (S6). Foam stability was calculated with the foam volume after 30 min of stirring as follows: Equation (S7).

#### 3.6.5. Swelling Index

To determine the swelling index, 7.5 mg of EPSwk were added in a falcon tube containing 15 mL of distilled water without homogenization, remaining at rest for 24 h. After this period, the solution was centrifuged for 5 min at 3300 rpm, and then the precipitate was weighed. The swelling index was calculated by Equation (S8).

#### 3.6.6. Film-Forming Ability

This analysis was performed according to the methodology of Davidović et al. [132], with adaptations. The film solutions were prepared at 1% (*w*/*v*), dissolving the powder of EPSwk in distilled water in a closed container, which was subjected to stirring and heating in a Digital Ultrasonic Vat until the material was completely solubilized. Then, the glycerol plasticizer was added to a concentration of 10 wt% (*w*/*w*), and film-forming solutions were cast in silicone molds and dried at 70 °C.

### 3.7. Photostability Study

This analysis was performed according to the methods of Melo et al. [133]. Initially, a 0.25% EPSwk (*w*/*v*) solution was prepared and placed in a borosilicate glass reactor to be irradiated with a 125 W Philips Mercury lamp without a bulb, used as a UV radiation source, under agitation (700 rpm), and temperature-controlled (25 ± 5 °C). The radiation intensity was 7.3 W, which was measured using a Hannar HI 97500 radiometer. For the evaluation of photostability, 1.0 mL aliquots of the solution were removed at pre-established time intervals and analyzed by a UV-Visible spectrophotometer (Cary 60 UV-Vis Spectrophotometer, Agilent Technologies, Santa Clara, CA, USA) in the 200–800 nm wavelength range. The absorption spectra of the samples were determined, and the degradation rate was calculated, from the wavelength of greatest absorbance, using Equation (S9).

A ball drop viscometer was used to assess the effects of UV radiation on intrinsic viscosity [134]. This analysis was performed at 0, 10, 30, 60, and 120 min of sample irradiation. The ball drop time in the solvent and irradiated samples were determined. The relative viscosity (ηrel) was calculated by the ball drop time of the sample solution divided by the ball drop time of the solvent. The specific viscosity (ηsp) and the intrinsic viscosity (η) were calculated using Equations (S10) and (S11), respectively.

### 3.8. Biological Assays

#### 3.8.1. Ex Vivo Mucoadhesiveness

This analysis was adapted from the methods of Abruzzo et al. [19]. A porcine buccal mucosa was used as a model due to its similarity with human oral tissue. The porcine mucosa was obtained from a local slaughterhouse. All processes were conducted by the ethical principles designated by the National Council for Animal Experimentation Control (CONCEA) and following the national legislation in force (Act No. 11,794, 8 August 2008, and Law No. 9605 of 12 February 1998). The mucosa was washed and kept in a phosphate buffer solution (PBS, pH = 7.4) and sent immediately to the laboratory. The test was accomplished on a TA-XT plus texture analyzer (TA Instruments, Surrey, UK) with a load cell of 5 kg. The test was performed on dry EPSwk (powder) and hydrogel (3% *w*/*v*) forms. Mucosa sections were fixed on specific support, which was kept in a water bath at 37 °C. A schematic diagram of the texture analyzer can be seen in Appendix A. Before each new test, the mucosa was previously moistened with 50 μL of simulated saliva solution (pH = 6.8) at 37 °C. The EPSwk powder was fixed with double-sided adhesive tape in the 10 mm P/10 probe. The EPSwk hydrogel was deposited (100 μL) on the mucosa fixed, and the sample was tested using the ½ ”P/0.5R probe. The tests were conducted at pre-test speeds of 0.10 mm s^−1^, test speeds of 0.50 mm s^−1^, and post-test speeds of 5.00 mm s^−1^, with a force of 0.2 N under the mucosa for 30 s. The Texture Analyzer (T.A. Exponent) software was employed to record the adhesion force (Fmax) and the area under the force versus distance curves (work of adhesion, Wad). Fmax (maximal detachment force) is the maximum force required to detach the tissue from the polymeric samples. The test was carried out in triplicate.

#### 3.8.2. Antimicrobial Activity

The broth dilution method was used to determine the minimum inhibitory concentration (MIC) of the EPSwk [135]. Initially, inoculants of the microorganisms *Escherichia coli* (CCBH3860) and *Staphylococcus aureus* (CCBH3856) were prepared by comparing the McFarland 0.5 tube scale (concentration equivalent to 1.5 × 10^8^ CFU mL^−1^). Subsequently, the initial inoculum was diluted to obtain a concentration of 1.5 × 10^5^ CFU mL^−1^, which was used to perform the test. Sample concentrations ranged from 78.125 to 10,000 µg mL^−1^. Two positive controls (inoculum + medium and inoculum + medium + solvent) and two negative controls (medium + solvent and solvent) were used to ensure the sterility of the medium as well as the viability, sensitivity, and sterility of the strains used. The test was performed in triplicate.

#### 3.8.3. In Vitro Cytotoxicity Assay

Cell culture conditions and treatments were performed according to the methods of Silva and Teixeira [136]. The human colorectal adenocarcinoma cell line (Caco-2) from the American Type Culture Collection (LGC Standards SLU, Barcelona, Spain) was used in this test. The EPSwk solution prepared in PBS (1000 µg mL^−1^) was sterilized in a membrane filter (pore size, 0.45 µm). This EPSwk solution was diluted, and the concentrations of 62.5, 125, 250, 500, and 1000 µg mL^−1^ were used in the test. Cells were plated in 48-well plates at a density of 25 000 cells/cm^2^. Cell viability was assessed 24 h after exposure of the samples by the MTT assay ([3-(4,5-dimethylthiazol-2yl)-2,5-diphenyl tetrazolium bromide]). The negative control was the vehicle (PBS) used to prepare the tested samples. Then, the cell culture supernatant was discarded, and the cells were incubated with 1 mL of MTT solution (0.5 mg mL^−1^ in supplemented DMEM) for 30 min at 37 °C. Then, the supernatant was eliminated, and 1 mL of DMSO was added to each well to solubilize the formed formazan crystals. The absorbance of the different solutions was assessed using a microplate spectrophotometer (Multiskan Ascent, Thermo Fisher Scientific, Waltham, MA, USA) at 570 nm. The MTT assay was carried out in a total of five experiments. The mean ± standard deviation of the percentage of cell viability was obtained using the OriginPro^®^ program.

#### 3.8.4. Hemocompatibility Assay

EPSwk was subjected to a hemolytic activity test on Sheep Blood Agar using the disk antibiogram technique, adapted to assess hemocompatibility [137]. Paper discs were cut and then sterilized in an autoclave. The disks were impregnated with 30 µL of the EPSwk solution, previously sterilized, at a concentration of 1% (1000 µg mL^−1^), and, after drying, they were placed in the Petri dishes containing the blood agar. For positive and negative control, the disks were impregnated, respectively, with 30 µL of Triton^®^ X-100 and 30 µL of distilled water (solvent used to prepare the EPS sample solution). After applying the discs, the plates were incubated at 35 ± 2 °C for 24 h. After incubation, the formation of hemolytic halos (measured in mm) was evaluated.

### 3.9. Microbiological Control and Water Activity (Aw)

To evaluate the presence or absence of Lactobacillus sp. in the EPSwk sample [138], the EPS solution was prepared at a concentration of 3% (*w*/*v*), and then serial dilutions up to 10^−3^ were performed. Aliquots of 0.1 mL of each dilution were transferred to plates containing MRS Agar using the surface scattering technique, and the plates were incubated at 37 ± 1 °C for 72 h under anaerobic conditions. The water activity (Aw) of the sample was determined at 25 °C using the portable water activity meter ETEC™. The test was performed in triplicate.

## 4. Statistical Analysis

Data were expressed as mean ± SD, and analysis of variance was performed with the OriginLab^®^ Program (Version 2020b). Differences determined were analyzed statistically by the Student’s t-test (*p* < 0.05).

## 5. Conclusions

The exopolysaccharide from water kefir (EPSwk) showed characterization of the typical homopolysaccharide dextran. EPSwk has thermal and photochemical stability compatible with exopolysaccharides. The biomaterial demonstrated potential for application as a stabilizer for emulsions, in addition to excellent mucoadhesive and film-forming properties. It also presented antimicrobial activity against *E. coli* and *S. aureus*. EPSwk exhibited good stability against microbiological contamination, non-cytotoxicity, and hemocompatibility, reaffirming the vast potential of this biopolymer for possible development of clean-label products aimed at the food, biomedical, cosmetic, and pharmaceutical areas.

## Figures and Tables

**Figure 1 molecules-27-03895-f001:**
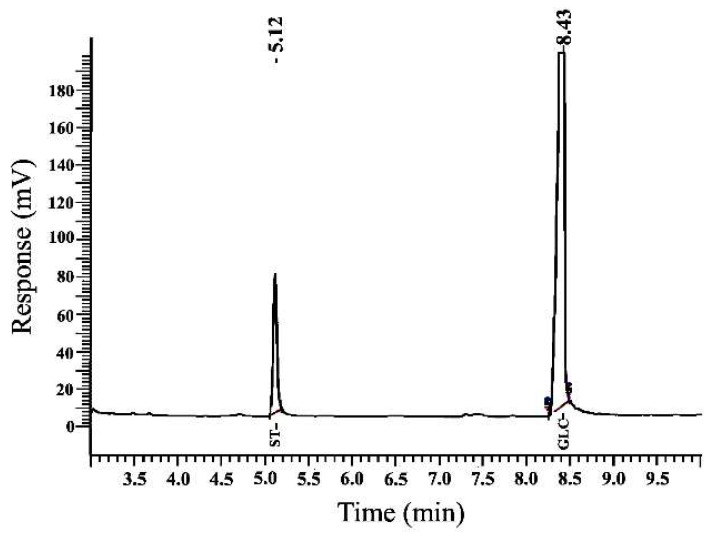
Chromatographic analysis by gas chromatography of an exopolysaccharide obtained from water kefir grains.

**Figure 2 molecules-27-03895-f002:**
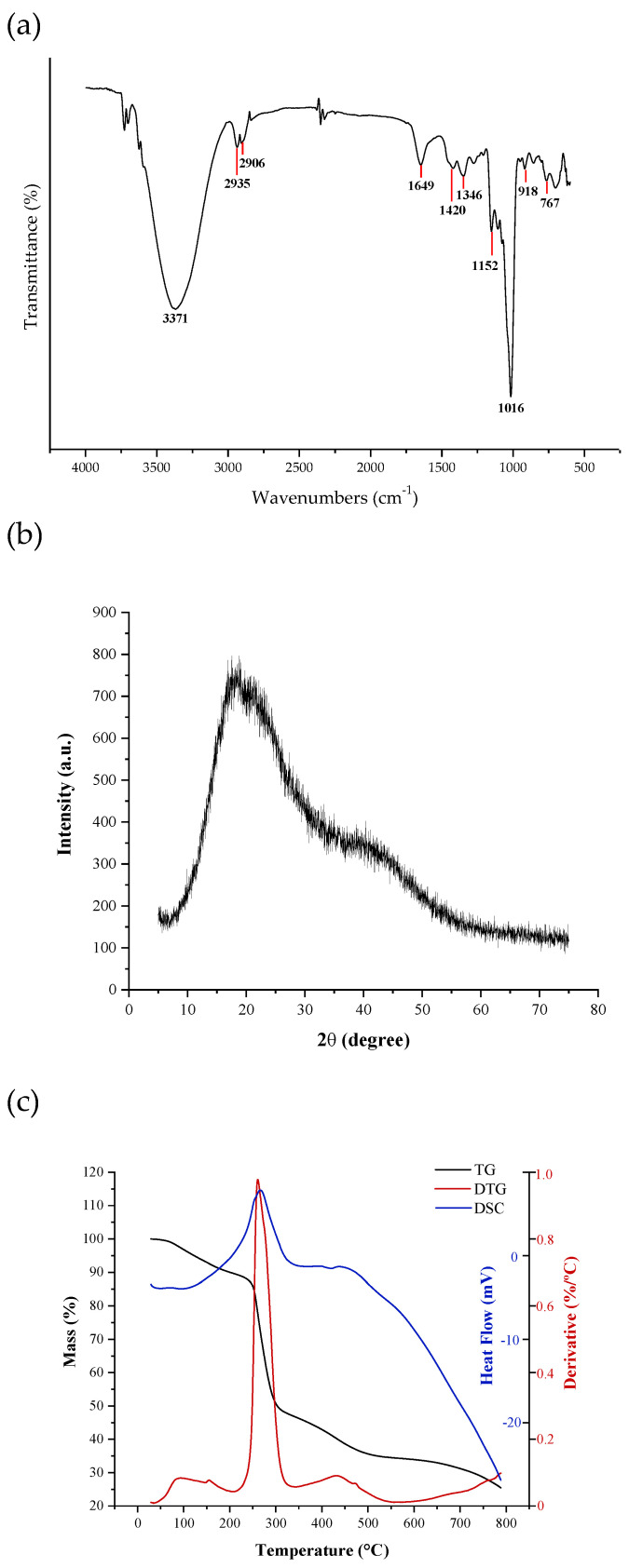
(**a**) FT-IR spectra, (**b**) X-ray diffractogram, and (**c**) TGA, DTG, and DSC curves of EPSwk.

**Figure 3 molecules-27-03895-f003:**
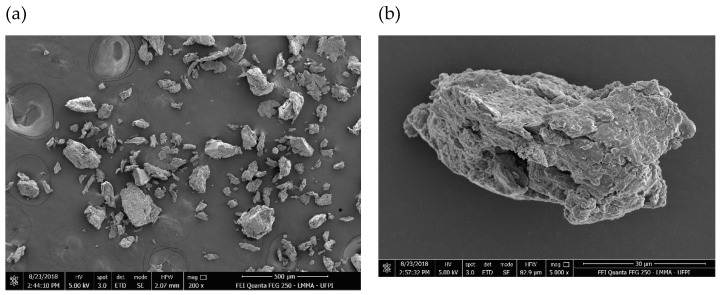
SEM image of EPSwk at (**a**) 200× and (**b**) 5000×.

**Figure 4 molecules-27-03895-f004:**
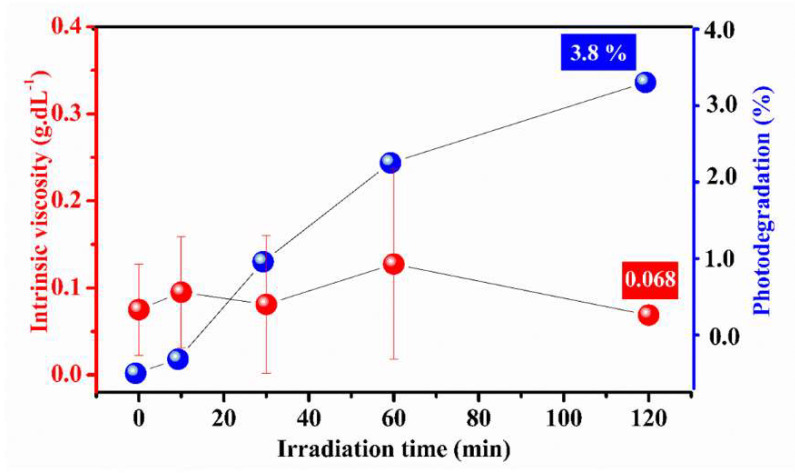
Intrinsic viscosity and photodegradation rate of EPSwk at different times of UV irradiation.

**Figure 5 molecules-27-03895-f005:**
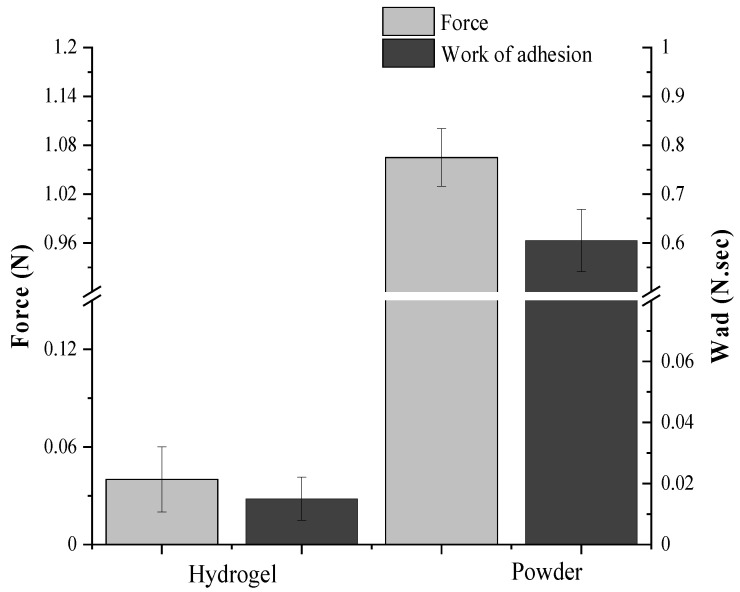
Mucoadhesiveness ex vivo of EPSwk samples.

**Figure 6 molecules-27-03895-f006:**
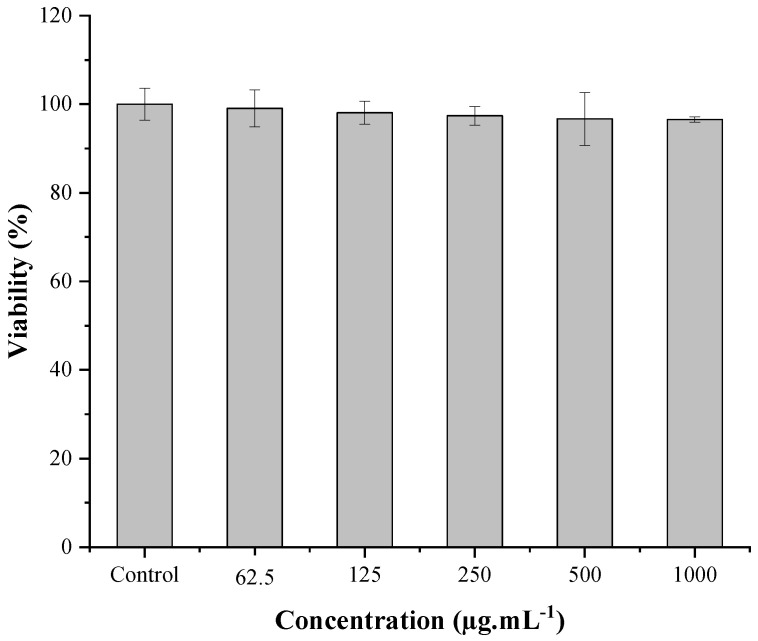
Cytotoxic evaluation of EPSwk in the Caco-2 cells by MTT reduction.

## Data Availability

The raw data required to reproduce these findings are available to download from (http://doi.org/10.17632/sycgz5m6dw.1). The processed data required to reproduce these findings are available to download from (http://doi.org/10.17632/sycgz5m6dw.1).

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
