# Peer review of "Biopolymer from Water Kefir as a Potential Clean-Label Ingredient for Health Applications: Evaluation of New Properties"

_molecules, 2022, doi:10.3390/molecules27123895_

Round 1

Reviewer 1 Report

Overall, this is a nice study, which contains a thorough analysis of a very up-to-date subject, which is application of green-derived biopolymers in various fields of science. The experimental methods are well selected. Albeit, at some points their analysis is superficial, and a too small population of samples is tested to allow for good statistical analysis (in some cases, only 2 samples – this should be corrected). Below is a detailed list points that require addressing before the study is to be published in a high-quality scientific journal.

  1. Line 27 there seems to be something wrong with the Mw unit – perhaps a superscript is missing? Maybe consider converting the Da to kDa?
  2. In the introduction, I think it would create a better impact and appeal to the broader audience if the Authors would provide more specific fields of applications. For example, when talking about biomedical, particular branches that could possibly employ such materials should be listed. Also, I think that a brief justification of the methods used for evaluating the particular properties should be given. It should also be stated why certain qualities are important.
  3. A brief discussion that would list possible (or already confirmed) disadvantages of this material and how to circumvent them would also strengthen the study.
  4. My suggestion would be to rename Appendix file to “supplementary materials”, so that the study is more in line with the instructions for Authors. Throughout the article, it should be indicated more clearly that the cited images and equations are to be found in the supplement. The current nomenclature (Eq. A1., Fig. A1, etc.) can be confusing as it differs only in one letter from the nomenclature of the data that is within the main body. I would suggest, for example: “Supplementary materials, Eq.1”.
  5. I think that for those who would whish to replicate the results of this study, it would be beneficial if the Authors provided the average (%) yield of the exopolysaccharide extraction.
  6. I would suggest using a simpler abbreviation of the material used. Although the Authors explain where the EPS-WKG name came from, for those who are just scamming the text, the presence of the hyphen in the name might suggest that this is in fact a composite or a co-polymer.
  7. Section 2.1 lacks any discussion with the literature – the obtained growth rates should be compared with other studies. Please, consider if showing the results as growth curves, giving the total population size, instead of bars with % of growth would not be more informative.
  8. Section 2.2 comparing the yield with different studies, the Authors should unify the units.
  9. Section 2.3, sentence at lines 103 – 105 is a pure tautology and should be rephrased so that it provides a valid explanation on how are the microorganisms affected by freezing. Description given in lines 117 – 124 needs correction so as to avoid unnecessary repetitions regarding the formation of ice crystals. Furthermore, this section needs discussion with the literature – were other studies similarly effective in protecting the kefir grains from cold-induced damage?
  10. Section 2.4
  11. Lines 138 – 140 too many significant digits are given for the electrical conductivity values. As pH analysis indicated presence of residual NaOH in the material, this should also be discussed in the aspect of the materials’ electrical conductivities. The obtained results should be compared with electrical properties of other polysaccharide solutions that can be found in literature. Finally, the Authors should try to suggest what is the charge carrier in their solutions.
  12. Lines 151 – 161. This section lacks a summary that would clarify if the obtained values of Zeta potential are as expected. Does Zeta potential affect the applicability of the material? And if so, how?
  13. Section 2.4.2 The Authors discuss how does the Mw value affect the performance and applicability of the materials. However, there is no information whether the Mw of the EPS-WKG obtained in this study is to be regarded as high or low.
  14. Section 2.4.3 Based on the fact that their material is a homopolysaccharide, composed of glucose units, the Authors indicate that their material is dextran. This is a far-fetched conclusion as there are many other homopolysaccharides composed solely of glucose (such as starch, pullulan, curdlan, etc.). This group of materials is referred to as glucans, wherein specific representatives of this family can be identified based on the type of O-glycosidic bond. Since the Authors did not analyze the type of this bond, they cannot identify which glucan representative they obtained (and it may as well be a mixture of more than one type).
  15. Section 2.4.5 The bands cannot be “intense”, they can only have high intensity. Furthermore, the region between 1900 – 1500 cm-1 is typical of double bonds, in particular 1650 cm-1 is usually assigned to C=O bonds, which are due to aldehyde functional groups in glucose. Hence, assigning these to O-H groups is rather incorrect. The quality of the spectrum presented is poor and it lacks an Y scale – this should be corrected. Finally, from the FTIR analysis one certainly cannot say that “the EPS-WKG corresponds to purified dextran”, the presented spectrum corresponds to any glucan.
  16. Section 2.4.6 stating that XRD assesses “organization” of the sample is rather imprecise and should be avoided in scientific literature. Lack of well-defined peaks in the X-ray diffractogram is rather an indicator of the amorphous structure (typical of polymers) than the “microcrystalline structure” and this should be corrected. My suggestion would be for the Authors to correct this section in accordance with the literature they have cited themselves. Again, the obtained results are not indicative of dextran.
  17. Section 2.4.7 Even though the Authors claim to have conducted their TG experiments in the oxidizing atmosphere of air, the residual mass of their material is 20% at 800oC – this almost impossible, as at such high temperature, the carbohydrates would completely combust, leaving no residues (as can be seen in the cited study [64: https://doi.org/10.1016/j.foodhyd.2018.10.053]). Hence, this result indicates that the study was rather conducted in an inert atmosphere. Since the Authors have used an STA device, my advice would be to analyze the DSC curve as it gives more information regarding the level of structural arrangement in the samples. When it comes to the analysis of the TG curves, the Authors compare their TG results with those obtained for the magnetite covered with dextran – this is a completely different material, so this is an obvious mistake. Finally, when discussing the importance of material’s thermal stability, one should list the typical processing temperatures, so that the reader can judge for himself if the obtained results are sufficient.
  18. Section 2.4.8 materials as such are not used “to control drug delivery systems” but may serve as drug delivery vehicles. Furthermore, the microstructural patterns as the ones presented herein should not be referred to as “fissures” as it is not a proper nomenclature from the materials science point of view.
  19. Section 2.5 the Authors should try to justify possible causes of the differences of the results they have obtained from the ones reported in the literature. Maybe this has something to do with the chain length, porosity or crystallinity? This section should be analyzed more thoroughly in respect to the results of the physicochemical analyses, presented in section 2.4.
  20. Section 2.6 What is the possible justification of the increased photostability of the tested material as compared to the ones reported in the literature?
  21. Section 2.7.1 Units need correction. I think that a schematic representation of the system employed to evaluate the mucoadhesiveness would be very helpful to understand what was actually evaluated and how. As methods are listed after the results, I think that the explanation what does the Fmax stand for should be given herein. The obtained values should be compared to other reports from literature and a brief estimation of the performance should be given.
  22. Line 302 – 306. The sentence needs rewriting.
  23. Line 344 “power” should be replaced with “powder”
  24. Line 356 There is no Fig. B5 – I believe this should read Fig. 5. And this figure needs correction as SD bars are unreadable right now.
  25. Section 2.7.2
  26. Line 366 units need correcting
  27. The Authors briefly state that other studies have also reported some antibacterial properties. I think that comparing the obtained MIC values would be of great importance. Of course, with some comments which would explain the differences (if there are any).
  28. Section 2.7.3
  29. Line 387 – units need correcting
  30. A brief justification of why the Caco-2 cells were selected for this study is required. If the intended application is oral mucoadhesives wouldn’t endothelial cells be more justified?
  31. Section 2.7.4 – the presented SEM images are insufficient to confirm the absence of microorganisms in the EPS. While I am not familiar with the Aw measuring technique for the estimation of the number of microorganisms in the sample, I believe that the obtained values should be compared with some standards or results obtained in different studies.
  32. Line 449 make and model of the centrifuge should be given
  33. Line 478 The following sentence needs rewriting to avoid repetition: “an evaporative detector with Agilent™ 1290 Infinity II Evaporative Light Scattering Detector (ELSD)”
  34. Line 495 Abbreviation EDTA is used without explanation what it stands for
  35. Line 501 it should be stated more clearly what sort of equipment the Agilent 7890B GC System is. And the abbreviation used should also be explained. While some people would know that GC stands for Gas Chromatography, for some this won’t be so obvious. And the study should be easily understandable to a broader audience, not necessarily experts in the analytical devices.
  36. Lines 506 – 511 For the FTIR analysis, it should be listed whether the spectra were recorded in the transmittance or absorbance mode. It should also be briefly explained how where the KBr pellets prepared (specifically, amount of KBr and the pressure used) and what served as a background. The following sentence is unclear to me: “The spectra of the exopolysaccharide samples were measured directly in the solid-state”. Could the Authors specify what they had in mind?
  37. Line 516 there seems to be something wrong with the unit
  38. Line 519 SDT Q600 V20.9 is not a thermogravimetric analyzer, it’s an STA device from the TA instruments company and this information should be specified. The amount of sample, type of crucible used, and a gas flow value should be listed. It is more typical to conduct the STA analyzes in the inert atmosphere – usage of an oxidizing atmosphere should be justified herein.
  39. Lines 521 – 528 Methodology of the SEM analysis needs rewriting as it contains errors (possibly due to loan translation). For example: “mounted on aluminum supports (stubs) with carbon tape, coated with a foil film” – what foil film do the Authors have in mind here? Stating that field emission was observed is a rather atypical description of SEM analysis. Furthermore, SEM does not have cannons.
  40. Lines 531-532 it is rather unacceptable to perform analyses as such on only two samples. From the point of view of statistical analysis, such analyses should be done in triplicate (at the very least). Only then, the homogeneity of the materials can be confirmed. Hence, I strongly recommend to increase the number of samples tested. Furthermore, as particular applications of the materials were not indicated, it should be justified why specific properties were evaluated and why are they referred to as functional.
  41. Line 534 it should be clearly indicated what solvent was used to prepare the EPS- WKG solution
  42. Line 541 WHC calculations lack some details as it is unclear what was used as a denominator in the equation. Also, it is somewhat unclear to me how is this parameter different from the swelling index. I mean, doesn’t it (roughly) estimate the same quality of the material? Could the Authors explain and justify evaluating both of these parameters? I would recommend either removing one of these or combining the two in one section.
  43. Line 546 Since the Authors claim to have prepared the w/v concentration, the final volume of the prepared mixture should be given and used as 100% in calculations (I expect this to be higher than the initial volume of water used).
  44. Line 548 more details on sample drying should be given
  45. Line 554 Shouldn’t the equation for calculating the foam stability use the initial foam volume as a denominator instead of a total suspension volume?
  46. Line 564 was the amount of glycerol added calculated in respect to the dry mass of the polysaccharide or to the total mass of the solution?
  47. Line 572 it should read “under magnetic agitation” and it should be specified how exactly was this employed.
  48. Lines 581 – 582 the sentence starting with “Initially (…)” needs rephrasing. From the method description it is unclear where did the specific and relative viscosity values come from.
  49. Lines 592 – 595 the sentences “Sections of the porcine buccal (…)” and “The mucosa was previously moistened (…)” needs rewriting ad they are grammatically incorrect.
  50. Lines 611 – 613 units need correction. 78.125 µg/ml is a rather unusual concentration, could the Authors justify its selection? It should be clarified what was used as a positive and negative control
  51. Lines 619 – 630 It should be clarified how exactly where the samples sterilized. It should also be clarified what was used as a solvent for preparing the solutions for tests. Different concentrations are selected for the cytocompatibility tests than for the antibacterial studies – why is that? The units of the concentrations need correction and the number of significant digits used to list the concentration should be unified throughout the study. The size of the cell well plates used for the study should be specified. And it should also be specified what served as a control. What was used to read the absorbances of test solutions?
  52. Lines 632 – 641. It appears that only one sample of each type was used in the hemocompatibility assay – this is unacceptable and at least three repetitions of the samples and the controls should be done. Again, concentration unit needs correction.
  53. Line 646 – unit needs correction. Again, analyses as such should be performed in triplicate.
  54. Lines 656 – 665. The Authors claim that “EPS-WKG showed characterization of the typical homopolysaccharide dextran”. However, in the study I don’t see any convincing proof that the material is in fact similar to dextran. In order to confirm that, analogous evaluation (FITR, XRD, TG) should be performed on dextran and the results should be compared. I would also recommend listing more specific applications.

Author Response

Dear reviewer, Thank you for your valuable and insightful contributions to our work. All comments were of significant importance for improving the study, and we tried to answer all questions in detail and attend to all possible suggestions. Please see the attachment.

Reviewer 2 Report

In the manuscript, Monalisa et al. extracted exopolysaccharide from water kefir grains, and studied the properties with various methods. The work is very interesting. However, there are some questions should be resolved before the manuscript is accepted for publication.

  1. In the manuscript, there is two appendixes, appendix A and appendix B. If the authors introduce appendixes in part 2, it will favor the readers.
  2. It is necessary for the authors to give specific sites and pH values of these sites.
  3. There are many abbreviations in the part of results, but, when they appear for the first, they were not given the full names.
  4. An optical image cannot prove good film formation, so it is suggested to add SEM, mechanical properties, WVP and other tests.
  5. Where is the SEM image(s)?

Author Response

(The authors gave the same response as above.)

Reviewer 3 Report

The paper “Biopolymer from water kefir as a potential clean-label ingredient for health applications: evaluation of new properties” by de Alencar Lucena has perfectly described a very novel subject with an excellent experimental design and valuable results. The following minor modifications should be applied to the text:

  1. There are some format errors in the text, please double-check the author’s guideline
  2. Some photos of the antimicrobial activity of the kefir EPS should be included in the text to have a better understanding and visualization.
  3. The importance of biopolymers should be properly addressed in the introduction part, the following paper is suggested to do so: https://doi.org/10.1080/10408398.2020.1843133
  4. Figure 2 should be inserted right after this section: 2.4.5 Fourier Transform Infrared Spectroscopy (FTIR)
  5. Couldn’t you have any control (commercial) sample to compare your results with?

Author Response

(The authors gave the same response as above.)

Round 2

Reviewer 1 Report

Overall, the authors did a good job correcting their study. Still, there are some minor issues that need to be addressed before the article is publishable.

Line 29 – unification in the EPSwk nomenclature is needed.

Lines 65 – 66 unnecessary repetition of the same information. And this needs rephrasing as α-glucan exopolysaccharide cannot be “constituted of dextran”, dextran is a glucan. And I believe that this not the only polysaccharide in the kefir grains.

Line 70 “is” is missing before the “biodegradable” word

Line 85 past tense is used whereas the rest of the paragraph is written in present tense. This mistake is also repeated on multiple occasions throughout the study.

Line 90 – 92 This is an improper definition of functional properties and it cannot be found in the cited articles. Functional properties are properties that determine the material’s function or performance. For example, if the material is to be used as a dressing, some of its functional properties might be: antibacterial effect, water holding ability, facile peel off, etc.

Lines 92, 95 phrases “protection capacity” and “good mucoadhesive” sound incorrect and need rephrasing

Lines 98 – 101 – there is a mix-up of plural and singular forms and this needs correcting

Line 110, in the sentence “This indicates that the fermentation conditions used were favorable to maintain the viability of the grains and facilitate their growth” it should be clearly indicated that the authors mean the conditions in this study.

Line 129 There is no Figure A2

Line 149 There is no Figure B2

Some fragments of the text are marked in red, as if they should be removed, but are not strikethrough. Lines 181-189, 431-439, 489-491, 773-784. This needs double checking

Section 2.4.1. Physico-chemical parameters and Zeta Potential

What authors suggest is not a sufficient explanation for the measured pH values. The polysaccharide itself could not increase the pH value making it alkaline. Yes, the OH groups in its chain may become ionized in the presence of the basic solvent, but this reaction is reversible – when the base is no longer present, the OH groups are not ionized anymore. Hence, if the authors had indeed neutralized the solution, the pH would not be basic. In fact this explanation “hydroxyl groups in the polysaccharide framework are ionized at alkaline pH. Thus, the alkaline pH of the biomaterial can be due to the presence of hydroxyl ions in the EPSwk chain” is completely incorrect! Presence of residual NaOH, as suggested in the previous version of the study, is far more probable. Especially given the fact that the conductivity measurements also seem to confirm that. I also asked the authors to compare their conductivity values with the literature – they did so, but not giving any explanation to the why their material is even more than 200 times more conductive at lower concentrations than the literature reports. Again, my guess would be residual NaOH.

Lines 226-228 Zeta potential is not directly correlated with the materials’ electrical conductivity, and the cited study does not claim that either! Zeta’s sign gives information about the particles’ overall surface charge, while its value gives information about the strength of electrostatic repulsion between the particles. The higher the absolute value, the more stable the suspension is. And it is generally assumed that stable suspension should have absolute values of Zeta potential higher than 25 mV. Hence, the values obtained by the authors rather suggest that their suspensions are not stable, and prone to flocculate and aggregate (opposite to what they’re claiming). This needs further discussion, as those are significant methodological and interpretational mistakes!

2.4.5. Fourier Transform Infrared Spectroscopy (FTIR)

One cannot state that presence of bands between 950 – 1200 cm^-1 indicates polysaccharides – every organic material has bands in this region, and they can be differentiated by the particular bands present in this fingerprint region. By the way, fingerprint region is generally acclaimed to be between 600 and 1400 cm^-1.

Now that the spectrum’s quality is slightly improved, I can see that it was subjected to some post-measurement treatment. This information should be clearly indicated in the text.

Section 2.7.2 Antimicrobial activity

The authors should clearly state if their MIC values are different from the literature (they are, in particular for the S. aureus) and try to explain differences. It would create a greater impact if the concentration units between this study and the cited studies were unified.

The authors state:  “we would like to apologize for not having the equipment available to analyze the DSC.” Yet, in their methods section one can read “The thermogravimetric analysis was performed using DSC-TGA equipment (TA Instruments, SDT Q600 V20.9, USA)”. And my previous suggestion that the Authors should add the DSC plot was based on the fact that they clearly are using an STA device, which measures TG and DSC simultaneously.

Author Response

Dear reviewer,

Please see the attachment. Thanks in advance for all your suggestions.
